# A Novel Central Camera Calibration Method Recording Point-to-Point Distortion for Vision-Based Human Activity Recognition

**DOI:** 10.3390/s22093524

**Published:** 2022-05-05

**Authors:** Ziyi Jin, Zhixue Li, Tianyuan Gan, Zuoming Fu, Chongan Zhang, Zhongyu He, Hong Zhang, Peng Wang, Jiquan Liu, Xuesong Ye

**Affiliations:** 1Biosensor National Special Laboratory, Key Laboratory of Biomedical Engineering of Ministry of Education, Zhejiang University, Hangzhou 310027, China; jinziyi@zju.edu.cn (Z.J.); gantianyuan@zju.edu.cn (T.G.); fzm21315045@zju.edu.cn (Z.F.); kevin_07@zju.edu.cn (C.Z.); jerryhe@zju.edu.cn (Z.H.); zhangh@mail.bme.zju.edu.cn (H.Z.); pengwangoptimus@zju.edu.cn (P.W.); 2College of Biomedical Engineering and Instrument Science, Zhejiang University, Hangzhou 310027, China; liujq@zju.edu.cn; 3Independent Researcher, 181 Gaojiao Road, Yuhang District, Hangzhou 311122, China; 21106107@zju.edu.cn

**Keywords:** camera calibration, point-to-point camera distortion calibration, vision-based human activity recognition, speckle pattern, digital image correlation

## Abstract

The camera is the main sensor of vison-based human activity recognition, and its high-precision calibration of distortion is an important prerequisite of the task. Current studies have shown that multi-parameter model methods achieve higher accuracy than traditional methods in the process of camera calibration. However, these methods need hundreds or even thousands of images to optimize the camera model, which limits their practical use. Here, we propose a novel point-to-point camera distortion calibration method that requires only dozens of images to get a dense distortion rectification map. We have designed an objective function based on deformation between the original images and the projection of reference images, which can eliminate the effect of distortion when optimizing camera parameters. Dense features between the original images and the projection of the reference images are calculated by digital image correlation (DIC). Experiments indicate that our method obtains a comparable result with the multi-parameter model method using a large number of pictures, and contributes a 28.5% improvement to the reprojection error over the polynomial distortion model.

## 1. Introduction

In recent years, vision-based human activity recognition (HAR) has developed rapidly with many exciting achievements [1,2,3]. Camera calibration is the upstream task of vision-based HAR, which can establish the mapping between real space and image space. Its accuracy determines the performance of downstream tasks such as feature points recognition and 3D reconstruction, and thereby affects the final performance of vision-based HAR [4]. For instance, the fisheye camera, which has been widely used in HAR tasks in the field of monitoring and security, although it has an ultra-wide-angle field of view, the object at the edge of the fisheye image has great deformation and serious information distortion. If the distortion of the camera is not accurately calibrated, it will seriously affect the accuracy of the subsequent algorithm. So, camera calibration is of great significance to vision-based HAR, containing daily activity recognition, self-training for sports exercises, gesture recognition and person tracking [5].

Distortion calibration of the camera impacts the accuracy of other parameters’ estimations. With the development of this field, distortion models’ degree of freedom is increasing, thus, there is much difference compared polynomial distortion models with point-to-point distortion models. In 1992, Weng [6] summarized distortion camera models, namely, radial, decentering, and thin prism distortions, which describe the real distribution of distortion by polynomials and parameters. Polynomial distortion models are idealized models and have a gap with the actual camera imaging relationship, resulting in limited accuracy of the calibration method. For higher accuracy of distortion calibration, some general distortion models and corresponding calibration methods [7,8,9,10,11,12,13] are proposed.

Since radial distortion is the main distortion of the camera, some researchers [7,8] developed a general radial distortion model that does not adopt a classical two-to-six parameter radial distortion, but rather a freer form of radial distortion. Inspired by their success, more general distortion models have been developed [9,10,11,12,13], describing lens distortion per pixel or by some kind of interpolation. In this kind of model, as distorted points can be extracted directly, the key problem to be solved is how to determine the original position (of pixels or spaces) of distorted points. Sagawa et al. employed structured-light patterns to obtain a dense distortion sample; the camera is aligned opposite to the target to make the feature points fixed [9]. Aubrey K. et al. set a synthetic image plane and recorded distortion as bias between real camera images and images projected on the synthetic image plane [10]. Jin et al. assumed that distortion in the central area of the image plane is negligible, and calculated distortion of the surrounding area by cross-ratio invariance [13]. Based on a raxel model, Thomas S. et al.’s pipeline [11] achieved the highest accuracy, but needs a large number of images. In our method, we designed a novel objective function that treats the distortion of each pixel as a constant quantity between different images and reprojects reference images by optimization results to create “virtual photos” which can determine the original position of distorted points.

Our method is based on the central generic camera model, which assumes all lights pass through a single optical center in the imaging process. Since the rays diverge from a point in the central generic camera model, the order and spacing ratio of rays remains unchanged, and the distortion rectification map remains unchanged with distance. Accordingly, there are sufficient reasons to believe that the distortion of a pixel is consistent across images taken with the same camera, which is the basis of our objective function.

Before the iteration, using the initial estimation of parameters with Zhang’s calibration method [14], we reprojected reference images to create “virtual photos” and extract dense features between “virtual photos” and original images. We designed our objective function to be a mean square error of the deformation between the “virtual photos” and original images. This objective function can remove the influence of distortion during parameter optimization, and obtain a more precise estimation of the camera parameters and target pose in each image.

To describe deformation adequately in the objective function, dense features are needed in our method. Although active phase targets can provide dense features [9,10,15,16], they are inconvenient to use. Chen et al.’s work [17] verified the accuracy and stability of feature detection methods based on digital image correlation (DIC). In Gao et al.’s work [18], the result of DIC is used to determine the accuracy of camera distortion calibration. Inspired by them, we incorporated a speckle pattern target and DIC into our camera calibration method, but unlike Chen, we did not utilize polynomial distortion models, but rather a full-pixel distortion description.

Since the polynomial distortion model is only an approximation of real distortion, the results of the camera calibration method based on the polynomial distortion model will be affected by incomplete distortion estimation. Our method can establish a point-to-point correspondence between distorted pixels and rectified pixels, which describes the camera distortion more comprehensively, and then gets a more accurate estimation of the camera parameters. Compared with methods based on the raxel model, our method needs only dozens of images, and strict experimental conditions are not required.

In our results, distortion is calculated for each point as the average value of the DIC calculation results across multiple images, which eventually formed a distortion rectification map that mapped images taken by the camera to undistorted ones. Figure 1 displays a distortion rectification map obtained by our point-to-point distortion calibration method. Figure 2 illustrates the difference between Figure 1 and the distortion rectification map obtained by Zhang’s method with a polynomial distortion model using the same set of calibration images, indicating free distortion, which the polynomial distortion model cannot describe.

The paper is organized as follows. Section 2 illustrates relative work. Section 3 introduces the camera model and lens distortion in our method. Section 4 describes our point-to-point distortion calibration method. In Section 5, experiments are performed to verify our method’s effectiveness. In Section 6, we discussed the issues not mentioned above. Finally, the conclusion is made in Section 7.

## 2. Related Work

### 2.1. Camera Model

From special to general, camera models can be classified as perspective cameras, central generic cameras, and non-central generic cameras [19]. The perspective camera is a single-view camera described by a pinhole imaging model, in which the imaging process is subjected to projective transformation, containing the finite projective camera and affine cameras [20].

The central generic camera contains the wide-angle camera, fisheye camera, and other cameras with refraction and reflection [19], which is unlikely to undergo a projective transformation and has a single focal point. In the imaging process of this camera, since the rays radiate from only one point, the order and spacing ratio of the rays remain unchanged, and the distortion rectification map remains constant with distance. That is why a distortion rectification map can describe the central generic camera’s distortion. Following distortion rectification, the central generic camera is simplified to be a camera that conforms to the pinhole imaging model.

The non-central generic camera is also referred to as a general camera. It lacks a single focal point, the order and spacing ratio of the rays will vary with distance, and the distortion rectification map cannot be used for distortion correction. Michael D. Grossberg and Shree K. Nayar from Columbia University first proposed a raxel model for a general camera [21], which uses a point *p* and a direction q to describe a ray entering the camera from the outside and colliding with the sensor. Subsequent works on general camera calibration have adopted the raxel model [11,19,22,23,24].

### 2.2. Pattern Design and Feature Detection

While a chessboard or circle pattern target is usually used in camera calibration, methods for improving feature detection precision have been proposed [25,26,27,28,29]. Ha, and Hyowon et al. discussed a triangle pattern target [30]. The intersection of three triangles can be approximated using a series of third-order polynomials as control points. An active phase target is also used for calibration [9,10,15,16], which provides more freedom for feature setting and de-focus situations. Chen et al. utilized speckle patterns and extracted feature points using the DIC method [17]. Experiments demonstrated that calibrating with a speckle pattern produces a smaller reprojection error than calibrating with a chessboard or circle pattern.

### 2.3. Digital Image Correlation Method

Digital image correlation (DIC), first proposed by researchers from the University of South Carolina [31], is a method for determining material deformation. In its application there are two kinds of DIC: (1) 2D-DIC, which is used for flat materials and requires the materials to remain flat during measurement; and (2) Stereo-DIC, which is used for three-dimensional materials and deformation, and can handle more variable situations.

The core objective of DIC algorithms is to match points of interest (POI) from the speckle pattern feature on the surface of materials in images, which usually consists of two main steps: (1) obtaining an initial guess and (2) iterative optimization. In the first step, there are methods such as correlation criteria [32,33], fast Fourier transform-based cross-correlation (FFT-CC) [34], and a scale-invariant feature transform (SIFT) [35] for a path-independent initial guess. For iterative optimization, Bruck HA et al. [36] proposed the forward additive Newton–Raphson (FA-NR) algorithm, which was later improved and widely used. As calculating the gradient and the Hessian matrix in optimization progress is a noticeable burden, one feasible approach is simplifying the Hessian matrix by making some assumptions, thereby converting it to a forward additive Gauss–Newton (FA-GN) algorithm. Pan, B. et al. introduced the (IC-GN) algorithm into the DIC [37], which maintains a constant Hessian matrix and can be pre-computed.

## 3. Model of Camera and Lens Distortion

A camera can be regarded as a mapping between a 3D world and a 2D image. Our method was developed to address the issue of central generic camera calibration. To describe this 3D–2D mapping, we combined a pinhole camera model and a point-to-point lens distortion model.

### 3.1. Pinhole Camera Model

In the pinhole camera model, point **Pw** in the 3D world was transformed into a point (u, v) in an image after transformation in Equation (1) [20]. **T** (Equation (2)) is a rigid body transformation from point **Pw** in the world coordinate system to point (X, Y, Z) in the camera coordinate system, using the rotation matrix **R** and translation matrix **t**. **A** (Equation (3)) is an inner parameter matrix that transforms the point in the image coordinate system (the normalized camera coordinate system) to point (u, v) in the pixel coordinate system, where fx and fy are focal lengths in pixels, and cx and cy are pixel coordinates of the principle point. To normalize the image plane, the formula is divided by Z.
(1)$$\left[\begin{array}{c} u\\ v\\ 1 \end{array}\right]=\frac{1}{Z}A\cdot d\left(T\cdot P_{w}\right)$$
(2)$$T=\left[\begin{array}{cc} R & t\\ 0 & 1 \end{array}\right]$$
(3)$$A=\left[\begin{array}{ccc} f_{x} & 0 & c_{x}\\ 0 & f_{y} & c_{y}\\ 0 & 0 & 1 \end{array}\right]$$

Distortion d in Equation (1) describes the geometric deformation arising from the optical imaging system. In Zhang’s method [14], distortion is employed on normalized image planes using polynomial representation [6]. However, in our method, for generality, distortion is defined as unknown mapping.

### 3.2. Point-to-Point Lens Distortion Model

This section will illustrate the generality of the point-to-point lens distortion model and its representation. Since **A** is a linear transformation, we can modify Equation (1) to apply distortion mapping on pixel coordinates.
(4)$$\left[\begin{array}{c} u\\ v\\ 1 \end{array}\right]=D\left(\frac{1}{Z}A\cdot T\cdot P_{w}\right)$$

By substituting D for d, the representation and rectification progress of distortion can be simplified. The distortion calibration result obtained with this lens distortion model can be shown as a point-to-point distortion rectification map. It can describe distortion caused by any central generic camera. If we rectify a central generic camera after obtaining point-to-point distortion rectification mapping, it is simplified to be a camera that conforms to the pinhole imaging model.

Figure 3 illustrates the mechanism of rectifying a camera with point-to-point distortion rectification mapping. Point-to-point mapping contains a mapping of the *X* direction and a mapping of the *Y* direction, which is stored as two matrices. Assuming a feature point is (u_de_, v_de_) in a deformed image, the corresponding point with the same feature in the reference image is point (u, v). An element (u, v) in the mapping matrix of the *X* direction stores the displacement $d_{u,v}^{x}$ of feature point (u_de_, v_de_)’s location in the deformed image relative to feature point (u, v)’s location in the reference image in the *X* direction. It is identical for the mapping matrix of the *Y* direction. Following that, the location of feature point (u_de_, v_de_) in the deformed image can be calculated using feature point (u, v)’s location in the reference image and element (u, v) in the mapping matrix of *X* and *Y* directions, as displayed in Equation (5).
(5)$$\begin{array}{c} u_{de}=u+d_{u,v}^{x}\\ v_{de}=v+d_{u,v}^{y} \end{array}$$

For every point (u, v) in the reference image, we can obtain its pixel value by copying the value of the corresponding point (u_de_, v_de_). If displacements $d_{u,v}^{x}$ and $d_{u,v}^{y}$ are decimals, bilinear interpolation is performed to obtain the value of the point (u_de_, v_de_). Going through every point (u, v) to obtain its value by Equation (5) and bilinear interpolation, a complete distortion corrected image is generated.

## 4. Method

Our method consists of three stages, which share the same set of calibration images. The first stage is an initial estimation. DIC method [38] is applied on images of speckle pattern calibration targets. Then, using Zhang’s approach, a set of control points extracted from DIC result is used for calibration. In the second stage, all parameters and distortions are optimized using a novel object function. In the third stage, distortion rectification mapping is extracted via point-to-point calculation. We will discuss each stage in detail in the following sections.

### 4.1. Initial Estimation

Our calibration target is based on a speckle pattern synthesized from Equation (6) [39]. In Equation (6), n and D are the number and radius (unit in pixels) of speckle, respectively. (x_k_, y_k_) is the random location of the kth speckle with a random peak intensity of $I_{k}^{0}$. The synthesized speckle pattern image is shown in Figure 4a, which is denoted as I^r^. We printed it as our calibration target. Additionally, we created a mask I^m^ with logical value representation, indicating the scope of the speckle pattern in image I^r^, as displayed in Figure 4b.
(6)$$I(x,y)=\sum_{k=1}^{n}I_{k}^{0}\exp\left[-\frac{{\left(x-x_{k}\right)}^{2}+{\left(y-y_{k}\right)}^{2}}{D^{2}}\right]$$

With the camera to be calibrated, we captured 15–30 images of this camera calibration target; the ith image is denoted as $I_{i}^{d}$. We allowed the speckle pattern area to extend beyond the photo’s edge. Figure 5 illustrates a calibration target’s pose in our calibration image. A rectangle outlines the image with thick solid lines. The array of black points represents control points for initial estimation. It is a noticeable principle that the speckled area can exceed the scope of the image, as shown on the left and bottom of Figure 6, but the array of control points must remain inside the scope of the image.

For initial estimation, we employ Zhang’s camera calibration method. Control points are extracted from the result of the DIC calculation performed on these images. DIC calculation can determine a point-to-point correspondence between points in reference and deformed images. The result of DIC calculation is expressed as displacement of pixels in the deformed image relative to corresponding pixels in the reference image. Equation (7) represents DIC calculation, where I^r^ is a reference image, I^m^ is the mask of I^r^, and $I_{i}^{d}$ is the deformed image. The displacement of all the pixels can be denoted as two mapping matrices, $M_{i}^{x}$ and $M_{i}^{y}$, corresponding to X and Y directions, respectively. If we have n deformed images, there are 2n mapping matrices.
(7)$$M_{i}^{x},M_{i}^{y}=F_{dic}(I^{r},I^{m},I_{i}^{d})$$

By using the DIC approach, we can obtain the displacement of pixels in the speckle pattern area of each $I_{i}^{d}$ relative to the corresponding point in I^r^ by the DIC method. We took displacement of an array of pixels in I^r^ and calculated their corresponding subpixel coordinates in a deformed image $I_{i}^{d}$ using displacement and pixel coordinates in I^r^, as in Equation (5). These corresponding points were saved as control points.

From initial estimation, we obtained camera parameter **A** and the pose of calibration targets **R****_i_** and **t****_i_**. Radial and tangential distortion is considered to obtain a more accurate calibration result.

### 4.2. Optimization with a Novel Objective Function

At this stage, we performed optimization with a novel objective function, Equations (11)–(13), that is also based on DIC. **A**, **R_i_**, and **t_i_** were used as optimization variables, with initial guess calculated by Zhang’s method. Then, we set radial and tangential distortion parameters to zero and reprojected reference image I^r^ with parameters **A**, **R_i_**, and **t_i_** to obtain “virtual photos” $P_{i}^{r}$, as in Equation (8). The mask I^m^ was also projected with the same method, as in Equation (9). Therefore, $P_{i}^{m}$ is a mask that indicates the scope of the speckle pattern in image $P_{i}^{r}$.
(8)$$P_{i}^{r}=\mathrm{Pr}oj\left(A,R_{i},t_{i},I^{r}\right)$$
(9)$$P_{i}^{m}=\mathrm{Pr}oj\left(A,R_{i},t_{i},I^{m}\right)$$

For every pixel in the projection of reference image $P_{i}^{r}$, the DIC method can obtain the displacement of the corresponding point in distorted image $I_{i}^{d}$ taken by the camera, as in Equation (10). A group of these, $P_{i}^{r}$, $P_{i}^{m}$ and $I_{i}^{d}$, is illustrated in Figure 6. It is worth noting that $P_{i}^{r}$ and $P_{i}^{m}$ share the same estimation of pose corresponding to $I_{i}^{d}$.
(10)$$M^{\prime }{}_{i}^{x},M^{\prime }{}_{i}^{y}=F_{dic}(P_{i}^{r},P_{i}^{m},I_{i}^{d})$$

Our objective function is set as the square error of these 2n mapping matrices, as in Equations (11)–(13). $\Delta p_{u,v,i}^{x}$ is element (u, v) of $M^{\prime }{}_{i}^{x}$, meaning *X* direction displacement of point (u, v) in image $I_{i}^{d}$ relative to the corresponding point in image $I_{i}^{proj}$. $\bar{\Delta p_{u,v}^{x}}$ is average of $\Delta p_{u,v,i}^{x}$ for every $\Delta p_{u,v,i}^{x}$ that does not equal to 0, $n_{x}$ is the number of $\Delta p_{u,v,i}^{x}$ that we took into account. $\Delta p_{u,v,i}^{y}$ and $\bar{\Delta p_{u,v}^{y}}$ are all the same for *Y* coordinate.
(11)$$\min\sum_{u,v}\sum_{i}^{n}\left[{\left(\Delta p_{u,v,i}^{x}-\bar{\Delta p_{u,v}^{x}}\right)}^{2}+{\left(\Delta p_{u,v,i}^{y}-\bar{\Delta p_{u,v}^{y}}\right)}^{2}\right],\;\forall \Delta p_{u,v,i}^{x}\neq 0,\Delta p_{u,v,i}^{y}\neq 0$$
(12)$$\forall \Delta p_{u,v,i}^{x}\neq 0,\Delta p_{u,v,i}^{y}\neq 0$$
(13)$$\bar{\Delta p_{u,v}^{y}}=\frac{\sum_{i}\Delta p_{u,v,i}^{y}}{n_{y}},\;\forall \Delta p_{u,v,i}^{y}\neq 0$$

This objective function means to minimize the difference of displacements between $P_{i}^{r}$ and $I_{i}^{d}$. When the optimization process was complete, we obtained new camera parameters and pose of calibration target, namely $A_{}^{o}$, $R_{i}^{o}$, and $t_{i}^{o}$.

### 4.3. Distortion Rectification Map Extraction

Distortion rectification maps of *X* and *Y* directions were calculated with reference image I^r^, calibration images $I_{i}^{d}$, and parameters $A_{}^{o}$, $R_{i}^{o}$, and $t_{i}^{o}$, obtained in optimization using a novel objective function.

Using parameters $A_{}^{o}$, $R_{i}^{o}$, and $t_{i}^{o}$, we reprojected reference image I^r^ and its mask I^m^ to obtain “virtual photos” $T_{i}^{r}$ and their mask $T_{i}^{m}$, as in Equations (14) and (15). DIC analysis was performed on every part of $T_{i}^{r}$, $T_{i}^{m}$, and $I_{i}^{d}$, as in Equation (16). $\Delta p_{u,v,i}^{o,x}$ is element (u, v) of $M_{i}^{o,x}$, and $\Delta p_{u,v,i}^{o,y}$ is element (u, v) of $M_{i}^{o,y}$. $\bar{\Delta p_{u,v}^{o,x}}$ is the average of every $\Delta p_{u,v,i}^{o,x}$ that does not equal 0, as in Equation (17). $n_{o,x}$ is the number of $\Delta p_{u,v,i}^{o,x}$ that we took into account.$\Delta p_{u,v,i}^{o,y}$ and $\bar{\Delta p_{u,v}^{o,y}}$ are all the same for *Y* coordinate, as in Equation (18). The result $\left(\bar{\Delta p_{u,v}^{o,y}},\bar{\Delta p_{u,v}^{o,y}}\right)$ is displacement $\left(d_{u,v}^{x},d_{u,v}^{y}\right)$ used in Equation (5). Therefore, point-to-point mapping of lens distortion is obtained.
(14)$$P_{i}^{o,r}=\mathrm{Pr}oj\left(A^{o},R_{i}^{o},t_{i}^{o},I^{r}\right)$$
(15)$$P_{i}^{o,m}=\mathrm{Pr}oj\left(A^{o},R_{i}^{o},t_{i}^{o},I^{m}\right)$$
(16)$$M_{i}^{o,x},M_{i}^{o,y}=F_{dic}\left(P_{i}^{o,r},T_{i}^{o,m},I_{i}^{d}\right)$$
(17)$$\bar{\Delta p_{u,v}^{o,x}}=\frac{\sum_{i}\Delta p_{u,v,i}^{o,x}}{n_{o,x}},\;\forall \Delta p_{u,v,i}^{o,x}\neq 0$$
(18)$$\bar{\Delta p_{u,v}^{o,y}}=\frac{\sum_{i}\Delta p_{u,v,i}^{o,y}}{n_{o,y}},\;\forall \Delta p_{u,v,i}^{o,y}\neq 0$$

## 5. Experiments

We conducted experiments to ascertain our method’s efficacy and priority. The convergence stability of our point-to-point distortion calibration method was proved by repeating experiments, which were repeated 10 times on 10 groups of images. Additionally, we evaluated the accuracy of the distortion rectification map calculated from the result of 7 training processes using a test set that was not used for the previous calibration. Additionally, the influence of the number of calibration images on the calibration results was investigated. We compared the performance of the distortion calibration results between our method, Zhang’s method [14], and Thomas S. et al.’s method [11], using 1920 × 1080 pixels laparoscopy, demonstrating a reprojection error and RMSE of camera parameter estimation. The ablation experiment demonstrated that optimization with a novel objective function and point-to-point calculation of lens distortion contributed to the final result’s improvement.

### 5.1. Experimental Procedures

The 2D targets employed in the experiments of Zhang’s method were circular and checkerboard pattern targets. We also adopted the deltille grid target proposed by Ha et al. [30] and the speckle pattern target proposed by Chen et al. [17]. As depicted in Figure 7a, the speckle pattern was synthesized using Equation (1) with n = 1.5 × 10^4^ and D = 60 pixels in a resolution of 4000 × 4000 pixels^2^. It was printed on adhesive matte paper by HP Indigo 7600 and stuck on a piece of glass to serve as a calibration target of 6 × 6 cm^2^. The circular pattern calibration target consisted of circulars with a 3 cm diameter and 6 cm center distance, forming a 7 × 7 array, as depicted in Figure 7b. The deltille grid pattern calibration target was composed of equilateral triangles with a side length of 6 cm and an arrangement, as demonstrated in Figure 7c. The checkerboard pattern calibration target had 6 × 6 cm^2^ squares, forming an 8 × 8 array, as in Figure 7d. The circular pattern, deltille grid pattern, and checkerboard pattern calibration targets were all printed on an alumina sheet with a glass substrate. To make a comparison under the same conditions, we used 7 × 7 array features extracted from the speckle pattern as the input of the method in [11]. Calibration images of each calibration target were captured by a 1920 × 1080-pixels binocular laparoscopy. We adjusted the lighting conditions to obtain the best imaging performance for each pattern, respectively, during image recording.

The experimental equipment was arranged as displayed in Figure 8. The calibration target was mounted on a mechanical arm, which was programmed to change its pose by inclination from −24° to 24° with a 6° interval. We positioned the calibration target initially in such a way that its projection covered the entire image area. The calculations were performed on a server with 256 CPUs and 512 GB of memory.

### 5.2. Validity under Different Initialization

To investigate our method’s performance for each kind of calibration target, we grouped 20 images of different poses. For this, the poses of selected images had to be various, and all selected images had to cover the whole field of view. Figure 9 displays the poses of a group of selected images. We selected 10 groups of images as a training set.

Here, we verified the stability of our optimization’s convergence under different initialization conditions by 10 training sets. The initial estimation was made by Zhang’s calibration method. Then, optimization using our novel objective function was performed, and a convergence curve was recorded. Figure 10 displays the average value and range of the convergence curve in 10 training processes. The vertical axis represents the value of the objective function described in Equations (11)–(13).

Additionally, we examined the distortion calibration results when different numbers of calibration images were utilized. For this purpose, the camera was calibrated 16 times, using from 10 to 40 calibration images. Then, training set images, undistorted by a distortion rectification map, were calibrated using Zhang’s calibration method, assuming no distortion remains. The reprojection error was recorded, indicating the accuracy of the distortion rectification map. Figure 11 illustrates the reprojection error of calibration using different numbers of calibration images. The reprojection error calculated from training results was smaller than the initial estimation, even when only 10 images were utilized, and remained stable when more than 20 images were used.

### 5.3. Ablation Study

Based on the initial estimation, we systematically added parts of our method and obtained a calibration result to demonstrate how individual parts influence the final performance. In the case of Map Extraction, the parameters obtained from the initial estimation were directly employed to calculate the point-to-point distortion rectification map, and calibration images were corrected by the point-to-point distortion rectification map. Then, assuming no distortion remained, the camera parameters were estimated using Zhang’s calibration method. Each configuration of the calibration progress was repeated with 5 groups of 20 images.

As listed in Table 1, the mean reprojection error of Map Extraction was reduced by 11.48%, compared to the result of the Initial Estimation. The last configuration contained our complete calibration progress, with a mean reprojection error reduction of 30.61% compared to the initial estimation result. The reprojection errors’ distribution in 5 repeated ablation experiments is showed in Figure 12. As a result, it can be inferred that in our method, both the optimizations with novel objective functions and the calculation of a point-to-point distortion rectification map are critical for improving calibration accuracy.

### 5.4. Benchmark Performance

This section compares the reprojection error and stability of the parameter estimation of previous methods with our novel method. Zhang’s calibration method with the circle pattern target, the checkerboard pattern target, the deltille grid target, proposed by Ha et al. [30], and the speckle pattern target proposed by Chen et al. [17] are included in the comparison. For Zhang’s method, using each target, we repeated the calibration progress 7 times using 7 groups of 20 pictures. A test set of 20 images was selected, excluding images in the training set. For the method of [11] and our method, we showed the reprojection error on the test set under the result of the calibration using 7 different groups of pictures. As to our method, for images in the test set, distortion was rectified using a point-to-point distortion rectification map calculated from the training result. Then, assuming no distortion remained, the camera parameters were estimated using Zhang’s calibration method.

The reprojection error is shown in Table 2. The method of the top 4 lines in Table 2 is Zhang’s calibration method with different calibration patterns. The reprojection errors of the chessboard, deltille grid, circle, and speckle calibration target methods were 0.34990613255, 0.115054 and 0.107224, respectively. Compared with Zhang’s calibration method using different targets, the reprojection error of our novel point-to-point distortion calibration method was the smallest as it was reduced by 28.5% beyond Zhang’s method using the same pattern.

The reprojection error was 0.075841 in the training result of our method, and was 0.076663 in the test result, exhibiting the performance of the distortion rectification map obtained from the training result on the new data. Although the reprojection error of the test set is slightly greater than that of the training set, it is still less than that of Zhang’s calibration approach with any type of calibration target. This demonstrates that the distortion correction map calculated from our point-to-point distortion calibration method could effectively correct new images captured with the same camera and achieve the desired impact.

To compare our point-to-point distortion calibration method with the method of [11], the performance on the test set under different amounts of calibration images is shown in Table 2 and Figure 13. Assuming that 20 images were used in our method, with the same number of images, the estimation result of [11] was inferior to that of our method because of overfitting, and when 228 images were used in [11], the estimation result was superior to that of our method with 20 images.

Table 2 and Figure 14 show the distributions of the internal parameters estimated using different calibration methods. The RMSE of the internal parameters’ estimation is listed in Table 2. The small circle in Figure 14 represents the average value of the estimated internal parameters, and the upper and lower sides of the error bar represent the max and min value of the estimated internal parameters, respectively. It can be inferred that with the method of Chen et al. [17] and our novel method, the internal parameters’ estimation in the repeated calibration is more stable than the other methods.

## 6. Discussion

We considered both the simulation and experimentation with a real camera when designing the experiment. In the simulation, the method employing the polynomial distortion model to simulate the camera distortion exhibits more advantages. In contrast, if we set additional distortions not limited to the polynomial distortion model, the point-to-point distortion calibration method offers more advantages. To make the experimental conditions neutral between the camera calibration method with the polynomial distortion model and point-to-point distortion calibration method, we used real cameras for our experiments.

As can be seen from the result of the validity experiment under a different initialization, the convergence curve of the optimization calculation by our method is stable, and the reprojection error is satisfactory when the number of calibration images involved in the optimization is not smaller than 20. The ablation study illustrated that the novel objective functions and the calculation of a point-to-point distortion rectification map have both resulted in a significant reduction of the reprojection error. The benchmark performance shows that the reprojection error of our method is smaller than that of methods using the polynomial distortion model. The accuracy of methods using the polynomial distortion model depends on whether the calibration pattern can achieve more accurate feature extraction and whether the features of image edges can be extracted. Our method not only uses the speckle pattern with higher feature extraction accuracy, but also adopts a full pixel distortion description and a specially designed objective function for optimization, so its reprojection error is superior to the method of the top 4 lines in Table 2.The method using the raxel model can achieve a smaller reprojection error than our method. When 228 images were used in the raxel model, the estimation result was superior to that of our method with 20 images. The following conclusion can be drawn from the findings of our experiment:(1)For our optimization process, over 20 calibration images can completely realize the objective function convergence.(2)Both the optimization with our new objective function and point-to-point distortion extraction significantly contributed to our method’s results.(3)The accuracy of our method is superior to methods using the polynomial distortion model. The method using the raxel model is more accurate, but with significantly more calibration images.

The setting of hyperparameters is a component in our technique that was not disclosed previously. To achieve the best performance of our method, hyperparameters were searched before calibrating different cameras in different environments. One of the hyperparameters was the subset size of the DIC calculation. The other one was the correlation coefficient threshold that determined which feature points were used in the parameter optimization.

1.Subset Size

In DIC, a larger size subset usually leads to a higher feature matching accuracy. However, an oversized subset introduces other problems, such as the complexity of deformation in the subset region. In this case, the current subset shape function cannot appropriately fit the subset deformation, resulting in decreased accuracy or failure of DIC. After a test with various subset sizes in our experiment, we used a subset with a radius of 70 pixels for DIC in the initial estimate and final verification, and a subset with a radius of 65 pixels in the parameter optimization.

2.Correlation Coefficient Cutoff

The correlation coefficient cutoff is used to determine whether the DIC results are reliable. A correlation coefficient cutoff that is set too high can introduce inaccurately matched features into the parameter optimization and reduce the accuracy of the parameter estimation. A correlation coefficient cutoff that is set too small results in large invalid regions of a calibration image that lack any features suitable for parameter optimization, which can also decrease the parameter estimation’s accuracy. After testing with different cutoff values, we used 0.065 as the cutoff value of the correlation coefficient in our experiment. This implies that features matched in the DIC with a correlation coefficient of less than 0.065 will be used for parameter optimization, whereas features matched in the DIC with a correlation coefficient of more than 0.065 will be filtered out.

Our method is devoted to the accurate calibration of camera parameters and lens distortion, which paves the way for a better performance of HAR. Developing Gao et al. [18] and Chen et al.’s work [17], our method can obtain a point-to-point distortion rectification map of the camera without establishing distortion models or strictly restricting experimental conditions.

## 7. Conclusions

We propose a camera calibration method that requires only dozens of images to obtain point-by-point distortion calibration results and internal camera parameters. This approach extracts dense features using a speckle pattern calibration target and DIC, as well as a new objective function for parameter optimization. The distortion rectification map is calculated from the result of the parameter optimization. We can warp camera-captured images into undistorted ones using a distortion rectification map. Compared with commonly used methods, this method is not limited to the polynomial distortion model, and also allows for the pixel-level calibration of the camera distortion. We designed experiments to validate our approach’s stability under various initialization conditions and compared it to the method of [11], using the same calibration target, and Zhang’s calibration method, utilizing a variety of calibration targets. Our method has a lower reprojection error than that of the compared method with the same number of calibration images, as demonstrated by experiments on a test set. This proves that our method can get a more accurate estimation of the camera distortion and camera parameters, so as to better describe the mapping between real space and image space. Therefore, our method is more advantageous than calibration methods using the polynomial distortion model in downstream tasks.

Despite the advantages above, our method is limited by its single optical center assumption, and its accuracy is inferior to that of methods using the raxel model. The accuracy of the distortion rectification map of our method is also limited by the number of images. As the DIC calculation at the edge of the speckle region is not accurate enough, there are some undesirable points that cannot be ruled out in the distortion rectification map. A possible solution is not to use pixels at the edges of the speckle region during distortion rectification map extraction. Another problem is computing the resource consumption of the DIC, which increases with the size of the subset area and the number of calibration images. This can be solved with GPU-accelerated computing [40]. These topics are on which we should concentrate our future efforts.

## Figures and Tables

**Figure 1 sensors-22-03524-f001:**
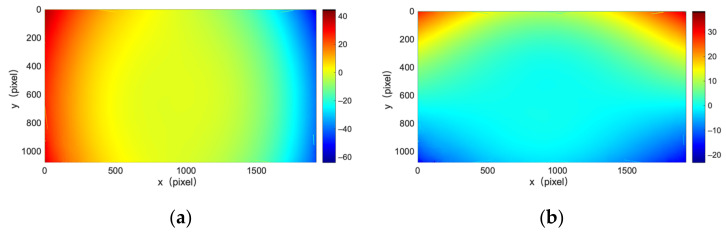
(**a**) Distortion rectification map of point-to-point calibration method for X directions; (**b**) Distortion rectification map of point-to-point calibration method for Y directions.

**Figure 2 sensors-22-03524-f002:**
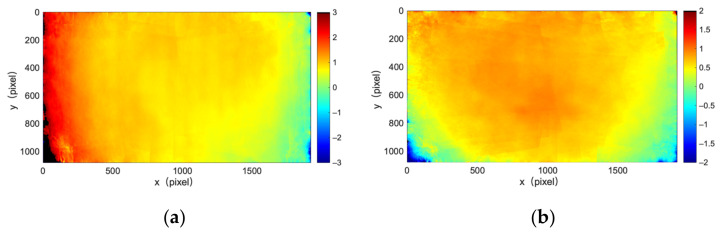
(**a**) Distortion rectification map of point-to-point calibration method subtracted from distortion rectification map of Zhang’s calibration method for X directions; (**b**) Distortion rectification map of point-to-point calibration method subtracted from distortion rectification map of Zhang’s calibration method for Y directions.

**Figure 3 sensors-22-03524-f003:**
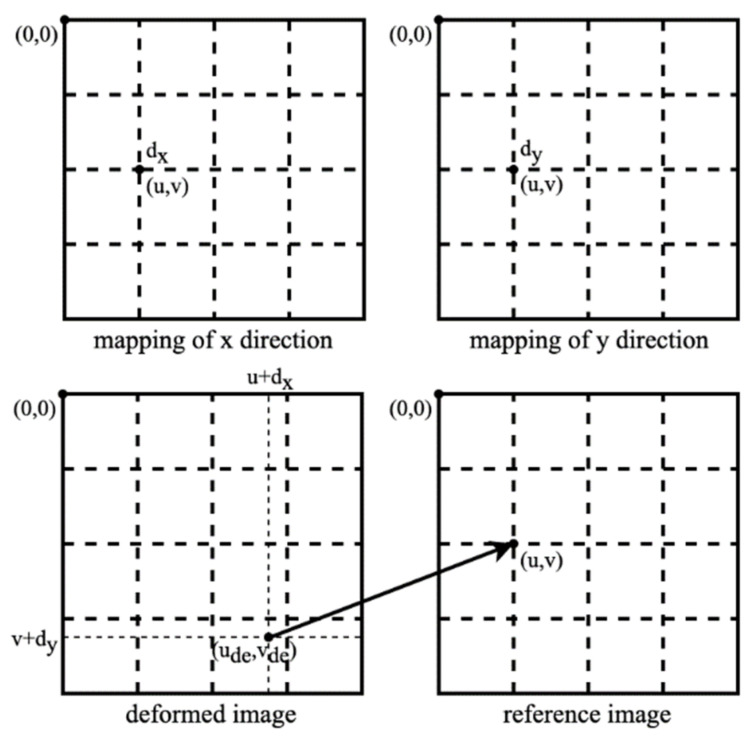
Mechanism of point-to-point mapping.

**Figure 4 sensors-22-03524-f004:**
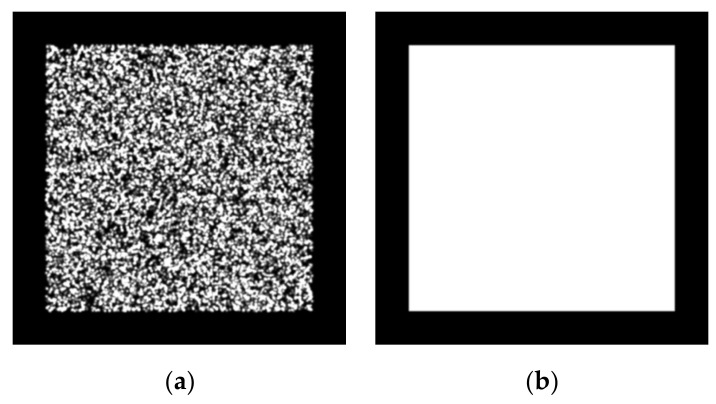
(**a**) Speckle pattern image; (**b**) Mask of speckle pattern image.

**Figure 5 sensors-22-03524-f005:**
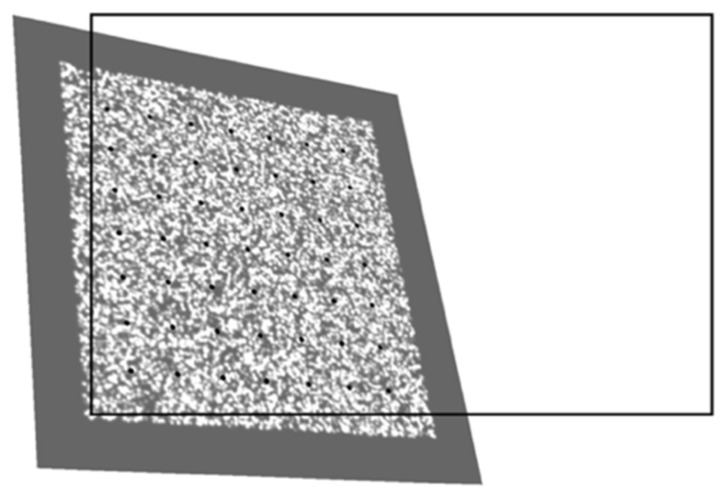
Taking an image of the speckle pattern calibration target.

**Figure 6 sensors-22-03524-f006:**
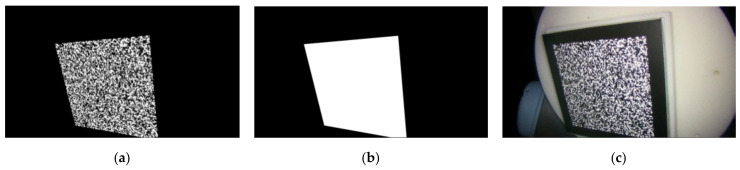
A group of images for DIC calculation in the second stage of our method, containing: (**a**) reprojection of reference image; (**b**) projection of mask; (**c**) image taken by the camera.

**Figure 7 sensors-22-03524-f007:**
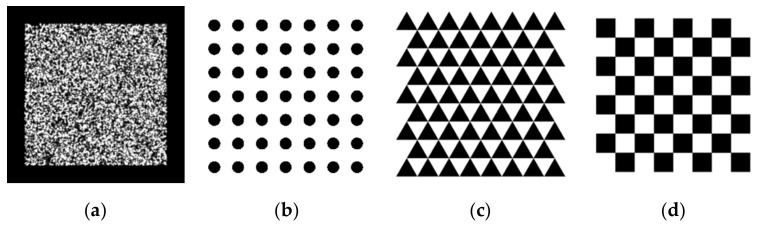
Two-dimensional targets used in the experiment, containing: (**a**) speckle pattern calibration target; (**b**) circular pattern calibration target; (**c**) triangle pattern calibration target; (**d**) chessboard pattern calibration target.

**Figure 8 sensors-22-03524-f008:**
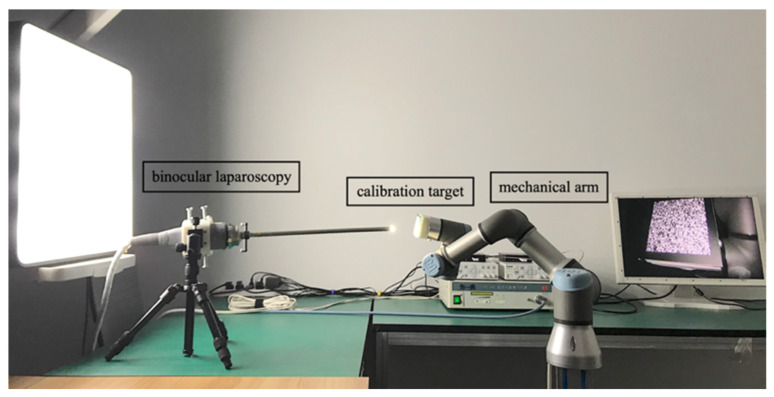
Experimental setup.

**Figure 9 sensors-22-03524-f009:**
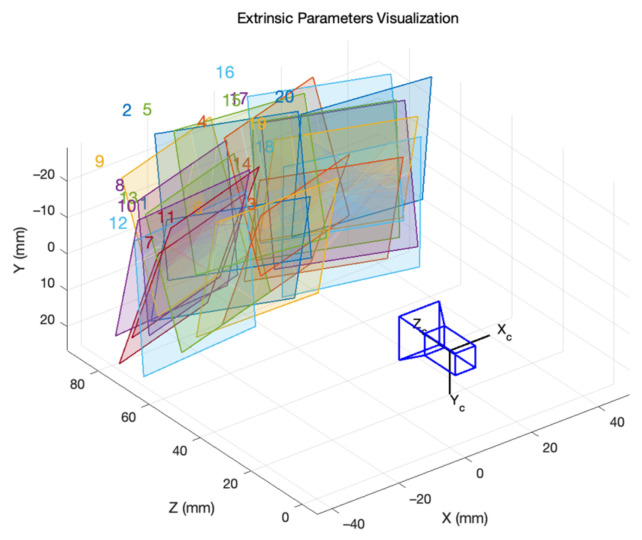
Poses of the target in a group of the training set.

**Figure 10 sensors-22-03524-f010:**
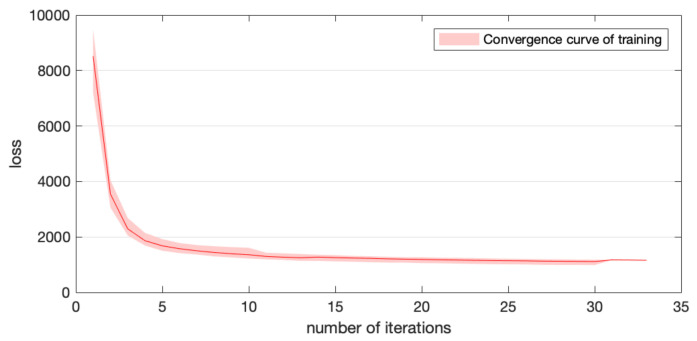
Average value and range of convergence curve in 10 training processes.

**Figure 11 sensors-22-03524-f011:**
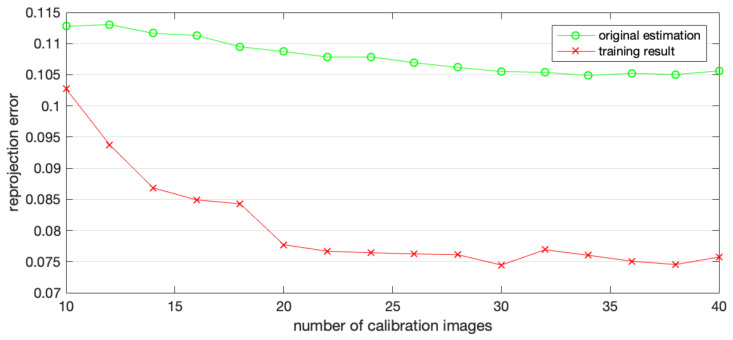
Reprojection error when different numbers of calibration images are used.

**Figure 12 sensors-22-03524-f012:**
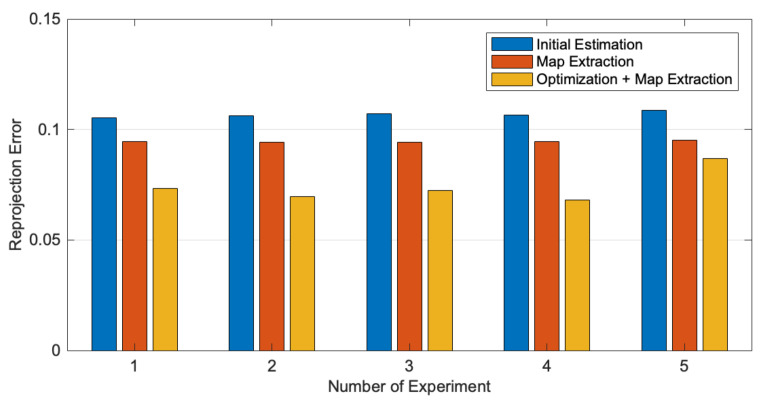
Reprojection errors’ distribution in ablation experiments.

**Figure 13 sensors-22-03524-f013:**
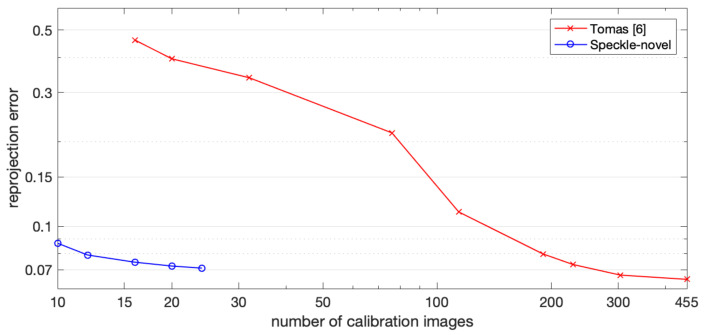
Reprojection error of Thomas S. et al.’s method and our point-to-point distortion calibration method on test set when different numbers of calibration images are used.

**Figure 14 sensors-22-03524-f014:**
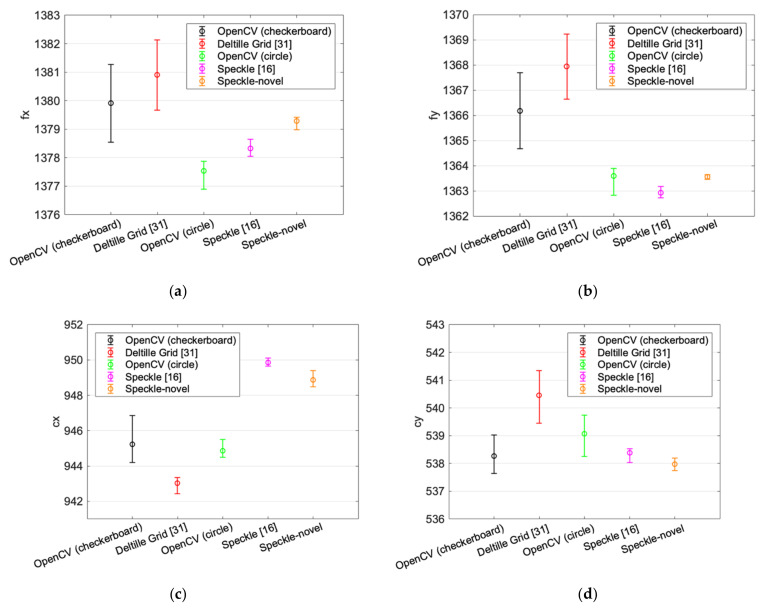
Distributions of internal parameters were estimated using different calibration methods. (**a**–**d**) are the distribution of estimated fx, fy, cx, and cy, respectively.

**Table 1 sensors-22-03524-t001:** The result of the ablation study.

Method	Mean Reprojection Error	Improvement (%)
Initial Estimation	0.106767	
Map Extraction	0.094512	11.48%
Map Extraction + Optimization	0.074087	30.61%

**Table 2 sensors-22-03524-t002:** Reprojection error and RMSE of internal parameters’ estimation of different calibration methods, with a training set of 228 images for method of line 6, and 20 images for other methods.

Method	Mean Reprojection Error	Root Mean Squared Error			
		Fx	Fy	Cx	Cy
OpenCV (checkerboard)	0.349906	1.04336	1.070489	0.839562	0.490924
Deltille Grid [30]	0.13255	0.788706	0.841719	0.298072	0.659631
OpenCV (circle)	0.115054	0.339109	0.391043	0.334438	0.484564
Speckle [17]	0.107224	0.221421	0.186815	0.168492	0.167473
Thomas [11]	0.352319	NA	NA	NA	NA
Thomas [11] (228 pic.)	0.072295	NA	NA	NA	NA
Speckle-novel	0.076663	0.14265	0.065153	0.292851	0.164638

## Data Availability

The source code, dataset, and result files are available at https://github.com/schcat, and accessed on 22 March 2022.

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
