# Peer review of "A Novel Central Camera Calibration Method Recording Point-to-Point Distortion for Vision-Based Human Activity Recognition"

_sensors, 2022, doi:10.3390/s22093524_

Round 1

Reviewer 1 Report

I think the article is very interesting but the discussion should be revised to clarify the results obtained and in the conclusions expand on the limitations of the study, future lines and improvements that can be implemented.

Reviewer 2 Report

The authors presented a calibration for human activity recognition that requires less data compared to the existing works. The calibration is not  only limited to the polynomial distortion model, but also allows for pixel-level calibration. The paper is generally well written and the method is sound. However there are several points to consider:

  • Please emphasize the research gap in the introduction section. Currently it is difficult to see the merit of the proposed approach in contrast to the existing ones.
  • There are two single-sentence paragraphs in section 2.1. Either develop the idea or merge them.
  • Authors don't need to put Thomas et al.'s method all the time in the body of the text. Citing the reference (as number) is sufficient.

Reviewer 3 Report

The paper presents an interesting subject, but the following aspects must be improved:

  • the title must be updated: it is not clearly why and how the calibration is correlated with human activity recognition? 
  • if this calibration is mapped for human activity recognition more detailed presentation must be added: what type of activities, what illumination conditions, etc
  • explain more clearly what are the novelty of the proposed method compared with other existing ones
  • comparison with other calibration methods must be added

Round 2

Reviewer 3 Report

Some of my comments were addressed. Still it is not clearly why this calibration is mapped for human activity recognition.

Please check equations: eg. 1 - 4, 12, 13, 17, 18.
